# Association of Toll-like receptor 7 variants with life-threatening COVID-19 disease in males: findings from a nested case-control study

Chiara Fallerini[1,2†], Sergio Daga[1,2†], Stefania Mantovani[3†], Elisa Benetti[2], Nicola Picchiotti[4,5], Daniela Francisci[6,7], Francesco Paciosi[6,7], Elisabetta Schiaroli[6], Margherita Baldassarri[1,2], Francesca Fava[1,2,8], Maria Palmieri[1,2], Serena Ludovisi[3,9], Francesco Castelli[10], Eugenia Quiros-Roldan[10], Massimo Vaghi[11], Stefano Rusconi[12,13], Matteo Siano[12], Maria Bandini[14], Ottavia Spiga[5,15], Katia Capitani[1,16], Simone Furini[2], Francesca Mari[1,2,8], GEN-COVID Multicenter Study[1], Alessandra Renieri[1,2,8]*, Mario U Mondelli[3,9], Elisa Frullanti[1,2]

[1]Medical Genetics, University of Siena, Siena, Italy; [2]Med Biotech Hub and Competence Center, Department of Medical Biotechnologies, University of Siena, Siena, Italy; [3]Division of Infectious Diseases and Immunology, Department of Medical Sciences and Infectious Diseases, Fondazione IRCCS Policlinico San Matteo, Pavia, Italy; [4]Department of Mathematics, University of Pavia, Pavia, Italy; [5]University of Siena, DIISM-SAILAB, Siena, Italy; [6]Infectious Diseases Clinic, Department of Medicine 2, Azienda Ospedaliera di Perugia and University of Perugia, Santa Maria Hospital, Perugia, Italy; [7]Infectious Diseases Clinic, "Santa Maria" Hospital, University of Perugia, Perugia, Italy; [8]Genetica Medica, Azienda Ospedaliero-Universitaria Senese, Siena, Italy; [9]Department of Internal Medicine and Therapeutics, University of Pavia, Pavia, Italy; [10]Department of Infectious and Tropical Diseases, University of Brescia and ASST Spedali Civili Hospital, Brescia, Italy; [11]Chirurgia Vascolare, Ospedale Maggiore di Crema, Crema, Italy; [12]Department of Biomedical and Clinical Sciences Luigi Sacco, University of Milan, Milan, Italy; [13]III Infectious Diseases Unit, ASST-FBF-Sacco, Milan, Italy; [14]Department of Preventive Medicine, Azienda USL Toscana Sud Est, Siena, Italy; [15]Department of Biotechnology, Chemistry and Pharmacy, University of Siena, Siena, Italy; [16]Molecular Mechanisms of Oncogenesis, ISPRO Core Research Laboratory (CRL), Firenze, Italy

*For correspondence: alessandra.renieri@unisi.it

†These authors contributed equally to this work

Group author details: GEN-COVID Multicenter Study See page 10

Competing interests: The authors declare that no competing interests exist.

## Abstract

**Background:** Recently, loss-of-function variants in TLR7 were identified in two families in which COVID-19 segregates like an X-linked recessive disorder environmentally conditioned by SARS-CoV-2. We investigated whether the two families represent the tip of the iceberg of a subset of COVID-19 male patients.

**Methods:** This is a nested case-control study in which we compared male participants with extreme phenotype selected from the Italian GEN-COVID cohort of SARS-CoV-2-infected participants (<60 y, 79 severe cases versus 77 control cases). We applied the LASSO Logistic Regression analysis, considering only rare variants on young male subsets with extreme phenotype, picking up TLR7 as the most important susceptibility gene.

**Results:** Overall, we found TLR7 deleterious variants in 2.1% of severely affected males and in none of the asymptomatic participants. The functional gene expression profile analysis demonstrated a reduction in TLR7-related gene expression in patients compared with controls demonstrating an impairment in type I and II IFN responses.

**Conclusions:** Young males with TLR7 loss-of-function variants and severe COVID-19 represent a subset of male patients contributing to disease susceptibility in up to 2% of severe COVID-19.

**Funding:** Funded by private donors for the Host Genetics Research Project, the Intesa San Paolo for 2020 charity fund, and the Host Genetics Initiative.

**Clinical trial number:** NCT04549831.

## Introduction

Coronavirus disease 2019 (COVID-19), a potentially severe systemic disease caused by coronavirus SARS-CoV-2, is characterized by a highly heterogeneous phenotypic presentation, with the large majority of infected individuals experiencing only mild or no symptoms. However, severe cases can rapidly evolve toward a critical respiratory distress syndrome and multiple organ failure (*Wu and McGoogan, 2020*). COVID-19 still represents an enormous challenge for the world's healthcare systems almost 1 year after the first appearance in December 2019 in Wuhan, Huanan, Hubei Province of China. Although older age and the presence of cardiovascular or metabolic comorbidities have been identified as risk factors predisposing to severe disease (*Hägg et al., 2020*), these factors alone do not fully explain differences in severity (*Stokes et al., 2020*). Stokes EK et al. reported that male patients show more severe clinical manifestations than females with a statistically significant (p<0.00001) higher prevalence of hospitalizations (16% versus 12%), ICU admissions (3% versus 2%), and deaths (6% versus 5%) (*Stokes et al., 2020*). These results are in line with other reports indicating that gender may influence disease outcome (*Garg et al., 2020*; *Goodman et al., 2020*).

These findings suggest a role of host predisposing genetic factors in the pathogenesis of the disease, which may be responsible for different clinical outcomes as a result of different antiviral defense mechanisms as well as specific receptor permissiveness to virus and immunogenicity.

Recent evidence suggests a fundamental role of interferon genes in modulating immunity to SARS-CoV-2; in particular, rare variants have recently been identified in the interferon type I pathway that are responsible for inborn errors of immunity in a small proportion of patients and auto-antibodies against type I interferon genes in up to 10% of severe COVID-19 cases (*Zhang et al., 2020*; *Bastard et al., 2020*).

Toll-like receptors (TLRs) are crucial components in the initiation of innate immune responses to a variety of pathogens, causing the production of pro-inflammatory cytokines (TNF-α, IL-1, and IL-6) and type I and II Interferons (IFNs), that are responsible for innate antiviral responses. In particular, the innate immunity is very sensitive in detecting potential pathogens, activating downstream signaling to induce transcription factors in the nucleus, promoting synthesis and release of type I and type II IFNs in addition to a number of other proinflammatory cytokines, and leading to a severe cytokine release syndrome which may be associated with a fatal outcome. Interestingly, among the different TLRs, TLR7 recognizes several single-stranded RNA viruses including SARS-CoV-2 (*Poulas et al., 2020*). We previously showed that another RNA virus, hepatitis C virus (HCV), is able to inhibit CD4 T cell function via Toll-like receptor 7 (TLR7) (*Mele et al., 2017*). Recently, *van der Made et al., 2020* have reported two independent families in which COVID-19 segregates like an X-linked recessive monogenic disorder conditioned by SARS-CoV-2 as an environmental factor.

Here, we performed a nested case-control study within our prospectively recruited GEN-COVID cohort with the aim to determine whether the two families described by van der Made et al. represent an ultra-rare situation or the tip of the iceberg of a larger subset of young male patients.

## Materials and methods

### Patients and samples

A subset of 156 young (<60 years) male COVID-19 patients was selected from the Italian GEN-COVID cohort of 1,178 SARS-CoV-2-infected participants (https://sites.google.com/dbm.unisi.it/gen-covid) (*Daga et al., 2021*). The study (GEN-COVID) was consistent with Institutional guidelines and approved

by the University Hospital (Azienda Ospedaliero-Universitaria Senese) Ethical Review Board, Siena, Italy (Prot n. 16929, dated March 16, 2020). We performed a nested case-control study (STREGA reporting guideline was used to support reporting of this study). Cases were selected according to the following inclusion criteria: i. male gender; ii. young age (<60 years); iii endotracheal intubation or CPAP/biPAP ventilation (79 participants). As controls, 77 participants were selected using the sole criterion of being oligo-asymptomatic not requiring hospitalization. Cases and controls represented the extreme phenotypic presentations of the GEN-COVID cohort. Exclusion criteria for both cases and controls were: i. SARS-CoV-2 infection not confirmed by PCR; ii. non-white ethnicity. Materials and methods details are listed in the Online Repository. A similar cohort from the second wave, composed of 83 young male COVID-19 patients, was used to expand the cohort.

## Statistical methods

We adopted the LASSO logistic regression, one of the most common Machine Learning algorithms for classification, that provides a feature selection method within the classification task able to enforce both the sparsity and the interpretability of the results (*Tibshirani, 1996*). In fact, the coefficients of the logistic regression model are directly related to the importance of the corresponding features, and LASSO regularization shrinks close to zero the coefficients of features that are not relevant in predicting the response, reducing overfitting and giving immediate interpretability of the model predictions in terms of few feature importance.

The principal components analysis (PCA) was applied prior to the LASSO logistic regression in order to remove samples that were clear outliers with respect to the first three principal components from the following analyses (deviating more than five standard deviations from the average).

A 10-fold cross-validation method was applied in order to test the performances. It provides the partition of the dataset into 10 batches, then nine batches are exploited for the training of the LASSO logistic regression and the remaining batch as a test, by repeating this procedure 10 times. The performance metrics are averaged on the 10 testing sets in order to avoid overfitting. The confusion matrix is built by summing up the predictions of the 10 testing folds. During the fitting procedure, the class unbalancing is tackled by penalizing the misclassification of the minority class with a multiplicative factor inversely proportional to the class frequencies.

In order to evaluate the significance of the association between *TLR7* variants and COVID severity, the Fisher's Exact Test was used.

For the quantitative PCR assay, the fold changes in mRNA expression level per gene were compared between the individual patients and controls using an unpaired t test on the log-transformed fold changes. p Values < 0.05 were considered statistically significant.

## In vitro peripheral blood mononuclear cell (PBMC) experiments

Peripheral blood mononuclear cells (PBMC) were isolated by Ficoll-Hypaque (GE Healthcare Bio-Sciences AB) density gradient centrifugation as previously described (*Mantovani et al., 2019*). $5 \times 10^5$ PBMC from COVID-19 patients 6 months after recovery and six unaffected male and female controls were stimulated for 4 hr with the TLR7 agonist imiquimod at 5 µg/mL or cell culture medium. Total RNA extraction was performed with RNeasy Plus Mini kit and gDNA eliminator mini spin columns (QIAGEN, Hilden, Germany), following the manufacturer's instructions. First-strand cDNA was synthesized from total RNA using High-Capacity cDNA Reverse Transcription Kit following the manufacturer's instructions (Thermo Fisher Scientific, Waltham, Massachusetts, United States). The Advanced Universal SYBR Green Supermix (BioRad, Redmond, WA, United States) was used. All reactions were performed in triplicates using the CFX96 Real-Time machine detection system (BioRad, Redmond, WA, United States) and each sample was amplified in duplicate. The following primers were used:

| | | |
|---|---|---|
| *TLR7* | Fw Primer | 5'-CATCAAGAGGCTGCAGATTAAA-3' |
| | Rv Primer | 5'-GAAAAGATGTTGTTGGCCTCA-3' |
| *IFN-γ* | Fw Primer | 5'-TGACCAGAGCATCCAAAAGA-3' |
| | Rv Primer | 5'-CTCTTCGACCTCGAAACAGC-3' |

*Continued on next page*

| IRF7 | Fw Primer | 5'-CCATCTTCGACTTCAGAGTCTTC-3' |
|---|---|---|
| | Rv Primer | 5'-TCTAGGTGCACTCGGCACAG-3' |
| ISG15 | Fw Primer | 5'-GACAAATGCGACGAACCTCT-3' |
| | Rv Primer | 5'-GAACAGGTCGTCCTGCACAC-3' |
| IFN-a | Fw Primer | 5'-GACTCCATCTTGGCTGTGA-3' |
| | Rv Primer | 5'-TGATTTCTGCTCTGACAACCT-3' |
| HRPT1 | Fw Primer | 5'-TGACACTGGCAAAACAATGCA-3' |
| | Rv Primer | 5'-GGTCCTTTTCACCAGCAAGCT-3' |

A total of 2.5 × $10^5$ PBMC from COVID-19 patients and healthy controls were maintained in RPMI-1640 supplemented with 10% of FCS, 1% antibiotic antimycotic solution, 1% L-glutamine and 1% Sodium Pyruvate (Sigma-Aldrich, St. Louis, MO, USA) and stimulated in vitro for 4 hr with Lipo-polysaccharide (LPS) at 1 µg/ml or cell culture medium and the Protein Transport Inhibitor GolgiStop (BD Biosciences, San Diego, CA, USA). After washing, PBMC were stained for surface cell marker using mouse anti-CD14PerCP-Cy5.5 (BD Biosciences) and anti-CD3BV605 (BD Biosciences) monoclonal antibody (mAb). Cells were fixed with BD Cytofix/Cytoperm and permeabilized with the BD Perm/Wash buffer (BD Biosciences) according to the manufacturer's instructions, in the presence of anti-IL6BV421 (BD Biosciences) mAb. Ex-vivo TLR7 intracellular expression was evaluated in PBMC from patients and controls by flow cytometry. 2,5 × $10^5$ PBMC were stained for surface markers using anti-CD19BV605, anti-CD14PerCP-Cy5.5 and anti-CD3BV421 (BD Biosciences) mAbs. Cells were fixed and permeabilized in the presence of anti-TLR7 Alexa Fluor 488 (R and D System, Minneapolis, MN, USA) mAb or isotype control as described above. After staining cells were washed, immediately fixed in CellFix solution (BD Biosciences) and analysed. Cell acquisition was performed on a 12-color FACSCelesta (BD Biosciences, San Diego, CA, USA) instrument. Data analysis was performed with the Kaluza 2.1 software (Beckman Coulter).

## Protein stability prediction

The protein structure of Human Toll Like Receptor, UniProtKB ID Q9NYK1 [https://www.uniprot.org/uniprot/Q9NYK1], was obtained by homology modeling using Swiss Model tool (*Waterhouse et al., 2018*). The selected template protein with 97% of sequence identity was the Crystal structure of monkey TLR7 with PDB ID 5GMF [https://www.rcsb.org/structure/5GMF]. The two Val to Asp missense mutations were analysed by using different protein stability predictors like Polyphen-2 (*Adzhubei et al., 2010*), SIFT (*Ng and Henikoff, 2003*), and DynaMut (*Rodrigues et al., 2018*).

## Transfection experiments of TLR7 variants

PCR based site-directed mutagenesis was performed in pUNO-hTLR7 plasmid (Invivogen), kindly provided by Ugo D'Oro (GSK Vaccines, Siena, Italy) (*Iavarone et al., 2011*), to generate specific plasmids for each *TLR7* variant, including those considered neutral (mutagenic primers available on request).

All point mutations except for p.Arg920Lys were confirmed by Sanger sequencing. HEK293 cells were maintained in DMEM supplemented with 10% FBS, 1% L-Glutamine and 1% penicillin/streptomycin at 37°C with 5% $CO_2$. Transient transfections were performed using Lipofectamine 2000 (Invitrogen) according to manufacturer's instructions: 3 × $10^5$ cell/well were seeded the day before, and then transfected with 2 µg of DNA. After 24 hr, the cells were stimulated with Imiquimod at 1 µg/ml for 4 hr and then total RNA was extracted with RNeasy Mini Kit (QIAGEN, Hilden, Germany). For each sample, cDNA was synthesized from 1 µg of total RNA using QantiTect Reverse Transcription kit (QIAGEN, Hilden, Germany) according to manufacturer's instructions. The expression of IFN-a in stimulated and unstimulated cells was evaluated by qRT-PCR using the same procedure as described for PBMCs.

## Results and discussion

We applied LASSO logistic regression analysis, after correcting for Principal Components, to a synthetic boolean representation of the entire set of genes of the X chromosome on the extreme phenotypic ends of the male subset of the Italian GEN-COVID cohort (https://sites.google.com/dbm.

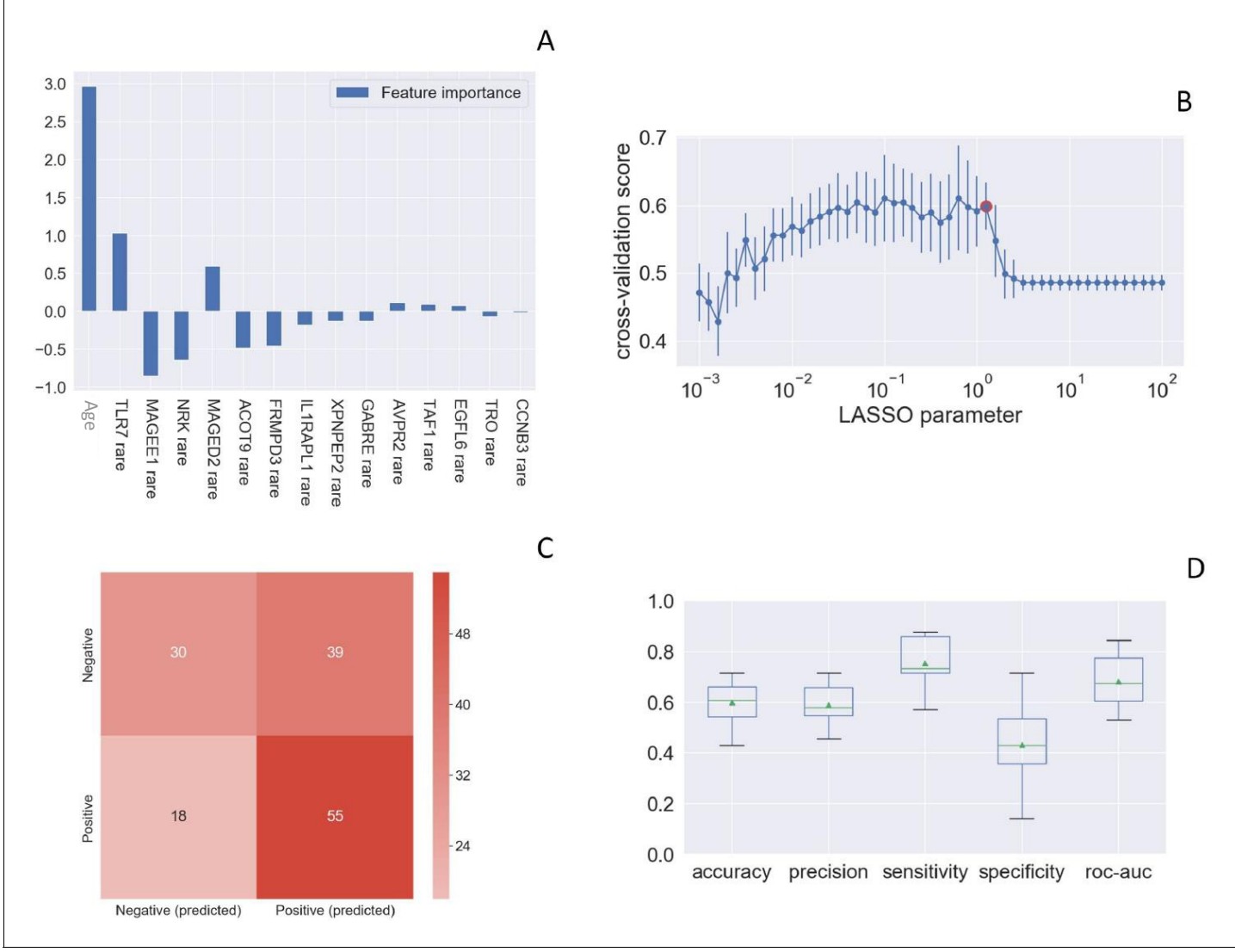

**Figure 1.** Rare *TLR7* variants and association with COVID-19. LASSO logistic regression on boolean representation of rare variants of all genes of the X chromosome is presented. *TLR7* is picked up by LASSO logistic regression as one of the most important genes on the X chr (Panel **A**). The LASSO logistic regression model provides an embedded feature selection method within the binary classification tasks (male patients with life-threatening COVID-19 vs infected asymptomatic male participants). The upward histograms (positive weights) reflect a susceptible behavior of the features to the target COVID-19, whereas the downward histograms (negative weights) a protective action. Panel **B** represents the cross-validation accuracy score for the grid of LASSO regularization parameters; the error bar is given by the standard deviation of the score within the 10 folds; the red circle (1.26) corresponds to the parameter chosen for the fitting procedure. Performances are evaluated through the confusion matrix of the aggregated predictions in the 10 folds of the cross-validation (Panel **C**) and with the boxplot (Panel **D**) of accuracy (60% average value), precision (59%), sensitivity (75%), specificity (43%), and ROC-AUC score (68%). The box extends from the Q1 to Q3 quartile, with a line at the median (Q2) and a triangle for the average.

**Table 1.** Fisher exact test of the overall combined cohorts in young males (<60 years).

| Clinical category | N. wild-type variants (97.84%) | N. pathological variants (2.15%) | Total |
|---|---|---|---|
| Severely affected males | 129 | 6 | 135 |
| Asymptomatic males | 104 | 0 | 104 |
| Total | 233 | 6 | 239 (Grand Total) |

p-value=0.0037.

**Table 2.** *TLR7* variants in severely affected Italian males -all ages- (cases).

| Nucleotide change | Amino acid change | dbSNP | CADD | ExAC_NFE | Function* | N. of patients | Clinical category† | Age | Cohort | Patient ID |
|---|---|---|---|---|---|---|---|---|---|---|
| c.901T>C | Ser301Pro | - | 26.4 | N/A | LOF | 1 | 3 | 46 | Italian | P3 |
| c.2759G>A | Arg920Lys | rs189681811 | 16.52 | 0.0002 | LOF‡ | 1 | 4 | 49 | Italian | P6 |
| c.3094G>A | Ala1032Thr | rs147244662 | 22.3 | 0.0006 | LOF | 2 | 3 | 65/66 | Italian | P7/P8 |
| c.655G>A | Val219Ile | rs149314023 | 12.28 | 0.0003 | HYPO | 1 | 4 | 32 | Italian | P1 |
| c.863C>T | Ala288Val | rs200146658 | 15.37 | 0.000012 | Neutral | 1 | 3 | 57 | Italian | P2 |
| c.1343C>T | Ala448Val | rs5743781 | 13.08 | 0.00465 | Neutral | 2 | 3 | 53/58 | Italian | P4/P5 |

CADD, Combined Annotation Dependent Depletion; ExAC, Exome Aggregation Consortium; NFE, Non-Finnish European;

*Function: HYPO, hypomorphic; LOF, loss-of-function;

†Clinical category: 4, Hospitalized and intubated; 3, Hospitalized and CPAP-BiPAP and high-flows oxygen treated; 2, Hospitalized and treated with conventional oxygen support only; 1, Hospitalized without respiratory support; 0, Not hospitalized oligo/asymptomatic individuals.

‡based on in silico prediction.

unisi.it/gen-covid) (*Daga et al., 2021*). The GEN-COVID study was consistent with Institutional guidelines and approved by the University Hospital (Azienda Ospedaliero-Universitaria Senese) Ethical Review Board, Siena, Italy (Prot n. 16929, dated March 16, 2020). Only rare variants ($\leq$1% in European Non-Finnish population) were considered in the boolean representation: the gene was set to one if it included at least a missense, splicing, or loss-of-function rare variant, and 0 otherwise. Fisher Exact test was then used for the specific data validation.

Toll-like receptor 7 (*TLR7*) was picked up as one of the most important susceptibility genes by LASSO Logistic Regression analysis (*Figure 1*). We then queried the COVID-19 section of the Network of Italian Genome (NIG) database (http://www.nig.cineca.it/, specifically, http://nigdb.cineca.it) that houses the entire GEN-COVID cohort represented by more than 1000 WES data of COVID-19 patients and SARS-CoV-2 infected asymptomatic participants (*Bastard et al., 2020*). By selecting for young (<60 year-old) males, we obtained rare (MAF $\leq$ 1%) *TLR7* missense variants predicted to impact on protein function (CADD > 12.28) in 5 out of 79 male patients (6.3%) with life-threatening COVID-19 (hospitalized intubated and hospitalized CPAP/BiPAP) and in none of the 77 SARS-CoV2 infected oligo-asymptomatic male participants.

We then investigated a similar cohort coming from the Italian second wave composed of male patients under 60 years of age without comorbidities (56 cases and 27 controls) was used to expand the cohort. All participants were white European. We found a *TLR7* variant in one of 56 cases (1.7%) and in none of 27 controls. Overall, the association between the presence of *TLR7* rare variants and severe COVID-19 was significant (p=0.037 by Fisher Exact test, *Table 1*).

We then investigated the presence of *TLR7* rare variants in the entire male cohort of 561 COVID-19 patients (261 cases and 300 controls) regardless of age. We found *TLR7* rare missense variants in three additional patients over 60 years of age, including two cases (who shared the p.Ala1032Thr variant) and one control (C1), bearing the p.Val222Asp variant, predicted to have a low impact on protein function (CADD of 5.36) (*Table 2*).

In order to functionally link the presence of the identified *TLR7* missense variants and the effect on the downstream type I IFN-signaling, we performed a gene expression profile analysis in peripheral blood mononuclear cells (PBMCs) isolated from patients following recovery, after stimulation with the TLR7 agonist imiquimod, as reported by *van der Made et al., 2020*. To explore all *TLR7* variants identified, we examined PBMCs from the control and all cases except P4 and P6 because them were not available. However, P4 and P5 shared the same variant. This analysis showed a statistically significant decrease of all *TLR7*-related genes for two variants (Ser301Pro and Ala1032Thr) identified in cases P3, P7, and P8 compared with healthy controls (Ctl) demonstrating a complete impairment of TLR7 signaling pathways in response to TLR7 stimulation (*Figure 2*, panel A and *Table 2*). The variant Val219Ile (P1) showed a hypomorphic effect determining a statistically

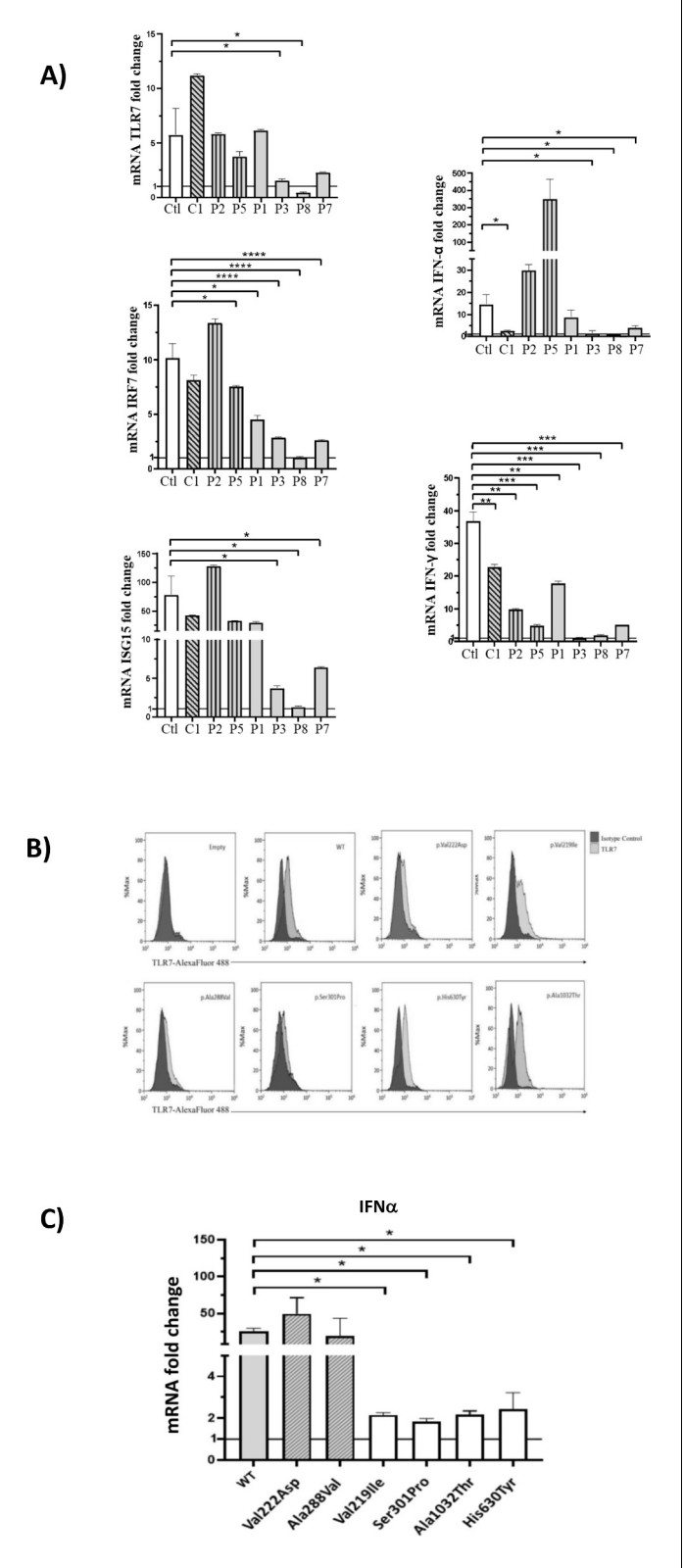

**Figure 2.** Gene expression profile analysis in peripheral blood mononuclear cells (PBMCs) and in HEK293 cells transfected with the functional variants after stimulation with a TLR7 agonist for 4 hr. (**A**) $5 \times 10^5$ PBMCs from COVID-19 patients and six unaffected male and female controls were stimulated for 4 hr with the TLR7 agonist imiquimod at 5 µg/mL or cell culture medium. Quantitative PCR assay was performed and the $2^{-\Delta\Delta Ct}$ calculated using *HPRT1* as housekeeping gene. Fold change in mRNA expression of *TLR7* and type 1 IFN-related genes *ISG15*, *IRF7*, *IFN-α* and *IFN-γ* induced by TLR7

*Figure 2 continued*

agonist imiquimod was compared with cell culture medium. Ctl indicates healthy controls (white bar); $C_1$, the asymptomatic mutated control (diagonal lines bar); P2, P5, cases with neutral variants (vertical lines bar); P1, P3, P8, P7 cases with functional variants (gray bar) (as in *Table 2*). (B) Histograms of intracellularly expressed TLR7 protein in HEK293 cells transfected with the different TLR7 plasmids. (C) Gene expression profile analysis of IFN-α in transfected cells after stimulation with the TLR7 agonist imiquimod. WT indicates cells transfected with WT TLR7 plasmid. Quantitative PCR assay was performed and the $2^{-\Delta\Delta Ct}$ calculated using HPRT1 as housekeeping gene. Fold change in mRNA expression induced by imiquimod was compared with cell culture medium. Error bars show standard deviation. p *values* were calculated for the reduction using an unpaired *t* test: *$p<0.05$; **$p<0.01$; ***$p<0.001$; ****$p<0.0001$.

significant decrease in mRNA levels only for IRF7 (directly activated by TLR7) and IFN-γ (*Figure 2*, panel A). Two Ala to Val variants identified in severely affected patients, Ala288Val and Ala448Val, were functionally neutral, that is not predicted to impair the TLR7 signaling pathways. This was confirmed by biochemical and structural analysis on the crystal structure of TLR7 protein (https://www.uniprot.org/uniprot/Q9NYK1). The prediction performed with different computational approaches showed both variants as benign with no effects on structural stabilization. Interestingly, the p. Val222Asp variant (C1) proved to be functionally neutral, in keeping with it being identified in the control and not in cases (*Figure 2*, panel A).

TLR7 expression was evaluated in monocytes and B cells from patients and healthy controls by flow cytometry. Patients and controls expressed the TLR7 protein at the intracellular level. The functional capacity of PBMCs was evaluated after stimulation with the TLR4 agonist lipopolysaccharide (LPS). Of note, LPS-induced production of IL6 by monocytes was similar in patients and controls (data not shown).

In order to validate the functional effect of *TLR7* variants, we have performed transfection experiments in HEK293 cells, cloning a dedicated TLR7 plasmid for each of them. Transfection experiments were performed in HEK293 cells that do not express endogenous TLR7 (*Chehadeh and Alkhabbaz, 2013*) and expression of TLR7 protein was examined by flow cytometry 24 hr after transfection, showing expression of TLR7 protein at the intracellular level in all cases (*Figure 2*, panel B). We then evaluated the expression of IFN-a in imiquimod stimulated and unstimulated cells by qRT-PCR employing the same assay described for PBMCs, confirming the results obtained in PBMCs for the screened variants (*Figure 2*, panel C).

Segregation analysis was available for two cases, P3 and P8 (*Figure 3*). In the two pedigrees, the disease nicely segregated as an X-linked disorder conditioned by environmental factors, that is SARS-CoV-2 (*Figure 3*, panel B). This was also supported by functional analysis on all *TLR7*-related genes (*Figure 3*, panel A). For example, expression profile analysis for *IRF7* gene in male mutated patient P8 confirmed a statistically significant reduction compared to the wild-type brother (*Figure 3*, panel A). Of note, only the infected mutated male had severe COVID-19, whereas the infected not mutated brother (II-2 of P8) was asymptomatic (*Figure 3*, panel C).

Our results showed that the two families reported by *van der Made et al., 2020*. with loss-of-function variants in males with severe COVID-19 with a mean age of 26 years represent a subset of COVID-19 male patients. Specifically, missense deleterious variants in the X-linked recessive *TLR7* gene may represent the cause of disease susceptibility to COVID-19 in up to 2% of severely affected young male cases (3/135, 2.2%). The same result was obtained for the entire male cohort, irrespective of age, with *TLR7* deleterious variants in 5/261 cases (1.9%). Since not all identified variants were functionally effective, the true percentage could be slightly lower in young males. Overall, males with rare missense variants shown here developed COVID-19 at a mean age of 56.5 years, considerably later than 26 years, in agreement with a predicted smaller impact on the protein than the loss of function deleterious variants reported by *van der Made et al., 2020*. Similarly, the identified rare missense *TLR7* variants impaired the mRNA expression of *TLR7* as well as the downstream pathway. The observation reported here may lead to consider *TLR7* screening in severely affected male patients in order to start personalized interferon treatment for those with this specific genetic disorder.

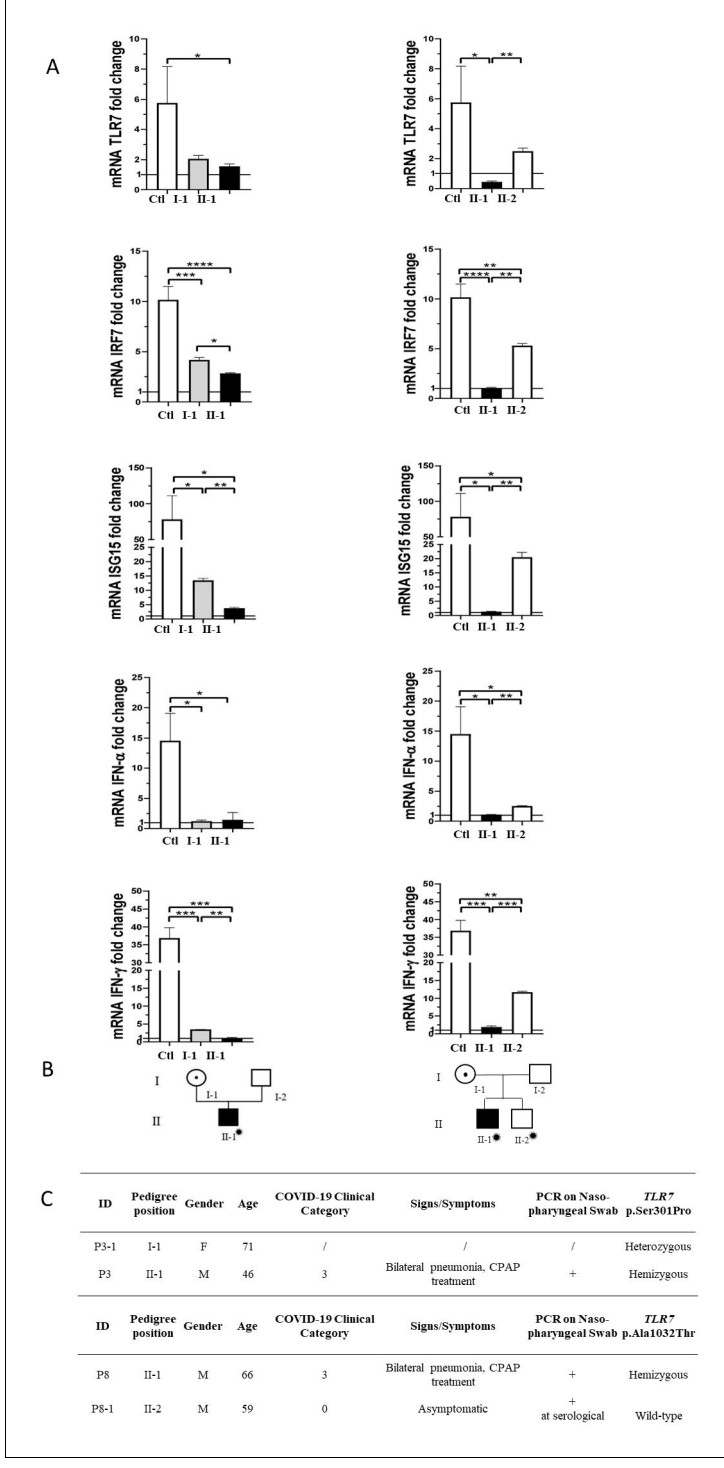

**Figure 3.** Segregation analysis. Fold change in mRNA expression following Imiquimod stimulation of *TLR7* itself and its main effectors, *IRF7*, *ISG15*, *IFN-alpha*, and *IFN-gamma* is shown in Panel **A**. Gray columns represent individuals harboring the TLR7 variant and black columns are severely affected SARS-CoV-2 cases. Pedigree (Panel **B**) and respective segregation of *TLR7* variant and COVID-19 status (Panel **C**) are also shown. Squares represent male family members; circles, females. Individuals infected by SARS-CoV-2 are indicated by a virus cartoon close to the individual symbol ( ).

## Acknowledgements

This study is part of the GEN-COVID Multicenter Study, https://sites.google.com/dbm.unisi.it/gen-covid, the Italian multicenter study aimed at identifying the COVID-19 host genetic bases. Specimens were provided by the COVID-19 Biobank of Siena, which is part of the Genetic Biobank of Siena, member of BBMRI-IT, of Telethon Network of Genetic Biobanks (project no. GTB18001), of EuroBioBank, and of RD-Connect. We thank the CINECA consortium for providing computational resources and the Network for Italian Genomes (NIG) http://www.nig.cineca.it for its support. We thank private donors for the support provided to AR (Department of Medical Biotechnologies, University of Siena) for the COVID-19 host genetics research project (D.L n.18 of March 17, 2020). We also thank the COVID-19 Host Genetics Initiative (https://www.covid19hg.org/), MIUR project 'Dipartimenti di Eccellenza 2018–2020' to the Department of Medical Biotechnologies University of Siena, Italy, and 'Bando Ricerca COVID-19 Toscana' project to Azienda Ospedaliero-Universitaria Senese. We also thank Intesa San Paolo for the 2020 charity fund dedicated to the project N B/2020/0119 'Identificazione delle basi genetiche determinanti la variabilità clinica della risposta a COVID-19 nella popolazione italiana'.

## Additional information

### Group author details

**GEN-COVID Multicenter Study**
Floriana Valentino: Medical Genetics, University of Siena, Siena, Italy; Med Biotech Hub and Competence Center, Department of Medical Biotechnologies, University of Siena, Siena, Italy; Gabriella Doddato: Medical Genetics, University of Siena, Siena, Italy; Med Biotech Hub and Competence Center, Department of Medical Biotechnologies, University of Siena, Siena, Italy; Annarita Giliberti: Medical Genetics, University of Siena, Siena, Italy; Med Biotech Hub and Competence Center, Department of Medical Biotechnologies, University of Siena, Siena, Italy; Rossella Tita: Genetica Medica, Azienda Ospedaliero-Universitaria Senese, Siena, Italy; Sara Amitrano: Genetica Medica, Azienda Ospedaliero-Universitaria Senese, Siena, Italy; Mirella Bruttini: Medical Genetics, University of Siena, Siena, Italy; Med Biotech Hub and Competence Center, Department of Medical Biotechnologies, University of Siena, Siena, Italy; Genetica Medica, Azienda Ospedaliero-Universitaria Senese, Siena, Italy; Susanna Croci: Medical Genetics, University of Siena, Siena, Italy; Med Biotech Hub and Competence Center, Department of Medical Biotechnologies, University of Siena, Siena, Italy; Ilaria Meloni: Medical Genetics, University of Siena, Siena, Italy; Med Biotech Hub and Competence Center, Department of Medical Biotechnologies, University of Siena, Siena, Italy; Maria Antonietta Mencarelli: Genetica Medica, Azienda Ospedaliero-Universitaria Senese, Siena, Italy; Caterina Lo Rizzo: Genetica Medica, Azienda Ospedaliero-Universitaria Senese, Siena, Italy; Anna Maria Pinto: Genetica Medica, Azienda Ospedaliero-Universitaria Senese, Siena, Italy; Laura Di Sarno: Medical Genetics, University of Siena, Siena, Italy; Med Biotech Hub and Competence Center, Department of Medical Biotechnologies, University of Siena, Siena, Italy; Giada Beligni: Medical Genetics, University of Siena, Siena, Italy; Med Biotech Hub and Competence Center, Department of Medical Biotechnologies, University of Siena, Siena, Italy; Andrea Tommasi: Medical Genetics, University of Siena, Siena, Italy; Med Biotech Hub and Competence Center, Department of Medical Biotechnologies, University of Siena, Siena, Italy; Genetica Medica, Azienda Ospedaliero-Universitaria Senese, Siena, Italy; Nicola Iuso: Medical Genetics, University of Siena, Siena, Italy; Med Biotech Hub and Competence Center, Department of Medical Biotechnologies, University of Siena, Siena, Italy; Francesca Montagnani: Med Biotech Hub and Competence Center, Department of Medical Biotechnologies, University of Siena, Siena, Italy; Dept of Specialized and Internal Medicine, Tropical and Infectious Diseases Unit, Azienda Ospedaliera Universitaria Senese, Siena, Italy; Massimiliano Fabbiani: Dept of Specialized and Internal Medicine, Tropical and Infectious Diseases Unit, Azienda Ospedaliera Universitaria Senese, Siena, Italy; Barbara Rossetti: Dept of Specialized and Internal Medicine, Tropical and Infectious Diseases Unit, Azienda Ospedaliera Universitaria Senese, Siena, Italy; Giacomo Zanelli: Med Biotech Hub and Competence Center, Department of Medical Biotechnologies, University

of Siena, Siena, Italy; Dept of Specialized and Internal Medicine, Tropical and Infectious Diseases Unit, Azienda Ospedaliera Universitaria Senese, Siena, Italy; Elena Bargagli: Unit of Respiratory Diseases and Lung Transplantation, Department of Internal and Specialist Medicine, University of Siena, Siena, Italy; Laura Bergantini: Unit of Respiratory Diseases and Lung Transplantation, Department of Internal and Specialist Medicine, University of Siena, Siena, Italy; Miriana D'Alessandro: Unit of Respiratory Diseases and Lung Transplantation, Department of Internal and Specialist Medicine, University of Siena, Siena, Italy; Paolo Cameli: Unit of Respiratory Diseases and Lung Transplantation, Department of Internal and Specialist Medicine, University of Siena, Siena, Italy; David Bennett: Unit of Respiratory Diseases and Lung Transplantation, Department of Internal and Specialist Medicine, University of Siena, Siena, Italy; Federico Anedda: Dept of Emergency and Urgency, Medicine, Surgery and Neurosciences, Unit of Intensive Care Medicine, Siena University Hospital, Siena, Italy; Simona Marcantonio: Dept of Emergency and Urgency, Medicine, Surgery and Neurosciences, Unit of Intensive Care Medicine, Siena University Hospital, Siena, Italy; Sabino Scolletta: Dept of Emergency and Urgency, Medicine, Surgery and Neurosciences, Unit of Intensive Care Medicine, Siena University Hospital, Siena, Italy; Federico Franchi: Dept of Emergency and Urgency, Medicine, Surgery and Neurosciences, Unit of Intensive Care Medicine, Siena University Hospital, Siena, Italy; Maria Antonietta Mazzei: Department of Medical, Surgical and Neuro Sciences and Radiological Sciences, Unit of Diagnostic Imaging, University of Siena, Siena, Italy; Susanna Guerrini: Department of Medical, Surgical and Neuro Sciences and Radiological Sciences, Unit of Diagnostic Imaging, University of Siena, Siena, Italy; Edoardo Conticini: Rheumatology Unit, Department of Medicine, Surgery and Neurosciences, University of Siena, Policlinico Le Scotte, Siena, Italy; Luca Cantarini: Rheumatology Unit, Department of Medicine, Surgery and Neurosciences, University of Siena, Policlinico Le Scotte, Siena, Italy; Bruno Frediani: Rheumatology Unit, Department of Medicine, Surgery and Neurosciences, University of Siena, Policlinico Le Scotte, Siena, Italy; Danilo Tacconi: Department of Specialized and Internal Medicine, Infectious Diseases Unit, San Donato Hospital Arezzo, San Donato Hospital Arezzo, Arezzo, Italy; Chiara Spertilli: Department of Specialized and Internal Medicine, Infectious Diseases Unit, San Donato Hospital Arezzo, San Donato Hospital Arezzo, Arezzo, Italy; Marco Feri: Dept of Emergency, Anesthesia Unit, San Donato Hospital, Arezzo, Italy; Alice Donati: Dept of Emergency, Anesthesia Unit, San Donato Hospital, Arezzo, Italy; Raffaele Scala: Department of Specialized and Internal Medicine, Pneumology Unit and UTIP, San Donato Hospital, Arezzo, Italy; Luca Guidelli: Department of Specialized and Internal Medicine, Pneumology Unit and UTIP, San Donato Hospital, Arezzo, Italy; Genni Spargi: Department of Emergency, Anesthesia Unit, Misericordia Hospital, Grosseto, Italy; Marta Corridi: Department of Emergency, Anesthesia Unit, Misericordia Hospital, Grosseto, Italy; Cesira Nencioni: Department of Specialized and Internal Medicine, Infectious Diseases Unit, Misericordia Hospital, Grosseto, Italy; Leonardo Croci: Department of Specialized and Internal Medicine, Infectious Diseases Unit, Misericordia Hospital, Grosseto, Italy; Gian Piero Caldarelli: Clinical Chemical Analysis Laboratory, Misericordia Hospital, Grosseto, Italy; Maurizio Spagnesi: Department of Preventive Medicine, Azienda USL Toscana Sud Est, Siena, Italy; Davide Romani: Department of Preventive Medicine, Azienda USL Toscana Sud Est, Siena, Italy; Paolo Piacentini: Department of Preventive Medicine, Azienda USL Toscana Sud Est, Siena, Italy; Elena Desanctis: Department of Preventive Medicine, Azienda USL Toscana Sud Est, Siena, Italy; Silvia Cappelli: Department of Preventive Medicine, Azienda USL Toscana Sud Est, Siena, Italy; Anna Canaccini: Territorial Scientific Technician Department, Azienda USL Toscana Sud Est, Siena, Italy; Agnese Verzuri: Territorial Scientific Technician Department, Azienda USL Toscana Sud Est, Siena, Italy; Valentina Anemoli: Territorial Scientific Technician Department, Azienda USL Toscana Sud Est, Siena, Italy; Agostino Ognibene: Clinical Chemical Analysis Laboratory, San Donato Hospital, Arezzo, Italy; Antonella D'Arminio Monforte: Department of Health Sciences, Clinic of Infectious Diseases, ASST Santi Paolo e Carlo, University of Milan, Milano, Italy; Federica Gaia Miraglia: Department of Health Sciences, Clinic of Infectious Diseases, ASST Santi Paolo e Carlo, University of Milan, Milano, Italy; Massimo Girardis: Department of Anesthesia and Intensive Care, University of Modena and Reggio Emilia, Modena, Italy; Sophie Venturelli: Department of Anesthesia and Intensive Care, University of Modena and Reggio Emilia, Modena, Italy; Stefano Busani: Department of Anesthesia and Intensive Care, University of Modena and Reggio Emilia, Modena, Italy; Andrea

Cossarizza: Department of Medical and Surgical Sciences for Children and Adults, University of Modena and Reggio Emilia, Modena, Italy; Andrea Antinori: HIV/AIDS Department, National Institute for Infectious Diseases, IRCCS, Lazzaro Spallanzani, Rome, Italy; Alessandra Vergori: HIV/AIDS Department, National Institute for Infectious Diseases, IRCCS, Lazzaro Spallanzani, Rome, Italy; Arianna Emiliozzi: HIV/AIDS Department, National Institute for Infectious Diseases, IRCCS, Lazzaro Spallanzani, Rome, Italy; Arianna Gabrieli: Department of Biomedical and Clinical Sciences Luigi Sacco, University of Milan, Milan, Italy; Agostino Riva: III Infectious Diseases Unit, ASST-FBF-Sacco, Milan, Italy; Department of Biomedical and Clinical Sciences Luigi Sacco, University of Milan, Milan, Italy; Pier Giorgio Scotton: Department of Infectious Diseases, Treviso Hospital, Local Health Unit 2 Marca Trevigiana, Treviso, Italy; Francesca Andretta: Department of Infectious Diseases, Treviso Hospital, Local Health Unit 2 Marca Trevigiana, Treviso, Italy; Sandro Panese: Clinical Infectious Diseases, Mestre Hospital, Venezia, Italy; Renzo Scaggiante: Infectious Diseases Clinic, ULSS1, Belluno, Italy; Francesca Gatti: Infectious Diseases Clinic, ULSS1, Belluno, Italy; Saverio Giuseppe Parisi: Department of Molecular Medicine, University of Padova, Padua, Italy; Stefano Baratti: Department of Molecular Medicine, University of Padova, Padua, Italy; Melania Degli Antoni: Department of Infectious and Tropical Diseases, University of Brescia and ASST Spedali Civili Hospital, Brescia, Italy; Matteo Della Monica: Medical Genetics and Laboratory of Medical Genetics Unit, A.O.R.N. "Antonio Cardarelli", Naples, Italy; Carmelo Piscopo: Medical Genetics and Laboratory of Medical Genetics Unit, A.O.R.N. "Antonio Cardarelli", Naples, Italy; Mario Capasso: Department of Molecular Medicine and Medical Biotechnology, University of Naples Federico II, Naples, Italy; CEINGE Biotecnologie Avanzate, Naples, Italy; IRCCS SDN, Naples, Italy; Roberta Russo: Department of Molecular Medicine and Medical Biotechnology, University of Naples Federico II, Naples, Italy; CEINGE Biotecnologie Avanzate, Naples, Italy; Immacolata Andolfo: Department of Molecular Medicine and Medical Biotechnology, University of Naples Federico II, Naples, Italy; CEINGE Biotecnologie Avanzate, Naples, Italy; Achille Iolascon: Department of Molecular Medicine and Medical Biotechnology, University of Naples Federico II, Naples, Italy; CEINGE Biotecnologie Avanzate, Naples, Italy; Giuseppe Fiorentino: Unit of Respiratory Physiopathology, AORN dei Colli, Monaldi Hospital, Naples, Italy; Massimo Carella: Division of Medical Genetics, Fondazione IRCCS Casa Sollievo della Sofferenza Hospital, San Giovanni Rotondo, San Giovanni Rotondo, Italy; Marco Castori: Division of Medical Genetics, Fondazione IRCCS Casa Sollievo della Sofferenza Hospital, San Giovanni Rotondo, San Giovanni Rotondo, Italy; Giuseppe Merla: Department of Molecular Medicine and Medical Biotechnology, University of Naples Federico II, Naples, Italy; Laboratory of Regulatory and Functional Genomics, Fondazione IRCCS Casa Sollievo della Sofferenza, San Giovanni Rotondo, Italy; Gabriella Maria Squeo: Laboratory of Regulatory and Functional Genomics, Fondazione IRCCS Casa Sollievo della Sofferenza, San Giovanni Rotondo, Italy; Filippo Aucella: Department of Medical Sciences, Fondazione IRCCS Casa Sollievo della Sofferenza Hospital, San Giovanni Rotondo, San Giovanni Rotondo, Italy; Pamela Raggi: Clinical Trial Office, Fondazione IRCCS Casa Sollievo della Sofferenza Hospital, San Giovanni Rotondo, San Giovanni Rotondo, Italy; Carmen Marciano: Clinical Trial Office, Fondazione IRCCS Casa Sollievo della Sofferenza Hospital, San Giovanni Rotondo, San Giovanni Rotondo, Italy; Rita Perna: Clinical Trial Office, Fondazione IRCCS Casa Sollievo della Sofferenza Hospital, San Giovanni Rotondo, San Giovanni Rotondo, Italy; Matteo Bassetti: Department of Health Sciences, University of Genova, Genova, Italy; Infectious Diseases Clinic, Policlinico San Martino Hospital, IRCCS for Cancer Research Genova, Genova, Italy; Antonio Di Biagio: Infectious Diseases Clinic, Policlinico San Martino Hospital, IRCCS for Cancer Research Genova, Genova, Italy; Maurizio Sanguinetti: Microbiology, Fondazione Policlinico Universitario Agostino Gemelli IRCCS, Catholic University of Medicine, Rome, Italy; Department of Laboratory Sciences and Infectious Diseases, Fondazione Policlinico Universitario A. Gemelli IRCCS, Rome, Italy; Luca Masucci: Microbiology, Fondazione Policlinico Universitario Agostino Gemelli IRCCS, Catholic University of Medicine, Rome, Italy; Department of Laboratory Sciences and Infectious Diseases, Fondazione Policlinico Universitario A. Gemelli IRCCS, Rome, Italy; Serafina Valente: Department of Cardiovascular Diseases, University of Siena, Siena, Italy; Marco Mandalà: Otolaryngology Unit, University of Siena, Siena, Italy; Alessia Giorli: Otolaryngology Unit, University of Siena, Siena, Italy; Lorenzo Salerni: Otolaryngology Unit, University of Siena, Siena, Italy; Patrizia Zucchi: Department of Internal Medicine, ASST Valtellina e Alto Lario, Sondrio, Italy; Pierpaolo

Parravicini: Department of Internal Medicine, ASST Valtellina e Alto Lario, Sondrio, Italy; Elisabetta Menatti: Study Coordinator Oncologia Medica e Ufficio Flussi Sondrio, Sondrio, Italy; Tullio Trotta: First Aid Department, Luigi Curto Hospital, Polla, Salerno, Italy; Ferdinando Giannattasio: First Aid Department, Luigi Curto Hospital, Polla, Salerno, Italy; Gabriella Coiro: First Aid Department, Luigi Curto Hospital, Polla, Salerno, Italy; Fabio Lena: Local Health Unit-Pharmaceutical Department of Grosseto, Toscana Sud Est Local Health Unit, Grosseto, Italy; Domenico A Coviello: U.O.C. Laboratorio di Genetica Umana, IRCCS Istituto G. Gaslini, Genova, Italy; Cristina Mussini: Infectious Diseases Clinics, University of Modena and Reggio Emilia, Modena, Italy; Giancarlo Bosio: Department of Respiratory Diseases, Azienda Ospedaliera di Cremona, Cremona, Italy; Enrico Martinelli: Department of Respiratory Diseases, Azienda Ospedaliera di Cremona, Cremona, Italy; Sandro Mancarella: U.O.C. Medicina, ASST Nord Milano, Ospedale Bassini, Cinisello Balsamo, Italy; Luisa Tavecchia: U.O.C. Medicina, ASST Nord Milano, Ospedale Bassini, Cinisello Balsamo, Italy; Marco Gori: Université Côte d'Azur, Inria, France; Lia Crotti: Istituto Auxologico Italiano, IRCCS, Department of Cardiovascular, Neural and Metabolic Sciences, San Luca Hospital, Milan, Italy; Department of Medicine and Surgery, University of Milano-Bicocca, Milan, Italy; Istituto Auxologico Italiano, IRCCS, Center for Cardiac Arrhythmias of Genetic Origin, Milan, Italy; Istituto Auxologico Italiano, IRCCS, Laboratory of Cardiovascular Genetics, Milan, Italy; Member of the European Reference Network for Rare, Low Prevalence and Complex Diseases of the Heart-ERN GUARD-Heart; Gianfranco Parati: Istituto Auxologico Italiano, IRCCS, Department of Cardiovascular, Neural and Metabolic Sciences, San Luca Hospital, Milan, Italy; Department of Medicine and Surgery, University of Milano-Bicocca, Milan, Italy; Chiara Gabbi: Independent Medical Scientist, Milan, Italy; Isabella Zanella: Department of Molecular and Translational Medicine, University of Brescia, Brescia, Italy; Clinical Chemistry Laboratory, Cytogenetics and Molecular Genetics Section, Diagnostic Department, ASST Spedali Civili di Brescia, Brescia, Italy; Marco Rizzi: Unit of Infectious Diseases, ASST Papa Giovanni XXIII Hospital, Bergamo, Italy; Franco Maggiolo: Unit of Infectious Diseases, ASST Papa Giovanni XXIII Hospital, Bergamo, Italy; Diego Ripamonti: Unit of Infectious Diseases, ASST Papa Giovanni XXIII Hospital, Bergamo, Italy; Tiziana Bachetti: Direzione Scientifica, Istituti Clinici Scientifici Maugeri IRCCS, Pavia, Italy; Maria Teresa La Rovere: Istituti Clinici Scientifici Maugeri IRCCS, Department of Cardiology, Institute of Montescano, Pavia, Italy; Simona Sarzi-Braga: Istituti Clinici Scientifici Maugeri, IRCCS, Department of Cardiac Rehabilitation, Institute of Tradate (VA), Tradate, Italy; Maurizio Bussotti: Istituti Clinici Scientifici Maugeri, IRCCS, Department of Cardiac Rehabilitation, Institute of Milan, Milan, Italy; Mario Chiariello: Istituto per lo Studio, la Prevenzione e la Rete Oncologica (ISPRO)-Core Research Laboratory and Consiglio Nazionale delle Ricerche-Istituto di Fisiologia Clinica, Siena, Italy; Mary Ann Belli: ASST Nord Milano, Ospedale Bassini, Cinisello Balsamo, Italy; Simona Dei: Health Management, Azienda USL Toscana Sudest, Tuscany, Italy

## Funding

| Funder | Grant reference number | Author |
|---|---|---|
| Private Donors for Host Genetics Research Project | D.L. n 18 of March 17 | Alessandra Renieri |
| Intesa San Paolo for 2020 charity fund | N.B.2020/0119 | Alessandra Renieri |
| Ministero dell'Istruzione, dell'Università e della Ricerca | Dipartimenti di Eccellenza 2018-2020 | Alessandra Renieri |
| Regione Toscana | Bando Ricerca COVID-19 Toscana | Alessandra Renieri |

The funders had no role in study design, data collection and interpretation, or the decision to submit the work for publication.

## Author contributions

Chiara Fallerini, Formal analysis, Writing - original draft; Sergio Daga, Margherita Baldassarri, Francesca Fava, Katia Capitani, Formal analysis, Methodology, Writing - original draft; Stefania Mantovani, Daniela Francisci, Francesco Paciosi, Elisabetta Schiaroli, Maria Palmieri, Serena Ludovisi,

Francesco Castelli, Eugenia Quiros-Roldan, Massimo Vaghi, Stefano Rusconi, Matteo Siano, Maria Bandini, Mario U Mondelli, Methodology, Writing - original draft; Elisa Benetti, Software, Formal analysis, Methodology, Writing - original draft; Nicola Picchiotti, Software, Methodology, Writing - original draft; Ottavia Spiga, GEN-COVID Multicenter Study, Data curation, Formal analysis, Methodology, Writing - original draft; Simone Furini, Data curation, Software, Formal analysis, Supervision, Validation, Methodology, Writing - original draft; Francesca Mari, Data curation, Methodology, Writing - original draft, Project administration; GEN-COVID Multicenter Study, Conceptualization, Methodology, Writing - original draft; Alessandra Renieri, Conceptualization, Data curation, Supervision, Writing - original draft, Project administration; Elisa Frullanti, Conceptualization, Data curation, Formal analysis, Supervision, Methodology, Writing - original draft, Project administration

### Author ORCIDs
Chiara Fallerini (iD) https://orcid.org/0000-0002-7386-3224
Sergio Daga (iD) https://orcid.org/0000-0002-6419-9456
Stefania Mantovani (iD) https://orcid.org/0000-0002-5885-2842
Elisa Benetti (iD) https://orcid.org/0000-0002-0819-604X
Nicola Picchiotti (iD) https://orcid.org/0000-0003-3454-7250
Margherita Baldassarri (iD) https://orcid.org/0000-0002-0391-1980
Francesca Fava (iD) https://orcid.org/0000-0002-4363-2353
Maria Palmieri (iD) https://orcid.org/0000-0003-3014-1552
Simone Furini (iD) https://orcid.org/0000-0002-1099-8279
Alessandra Renieri (iD) https://orcid.org/0000-0002-0846-9220
Elisa Frullanti (iD) https://orcid.org/0000-0001-5634-031X

### Ethics
Clinical trial registration NCT04549831.
Human subjects: The GEN-COVID study was consistent with Institutional guidelines and approved by the University Hospital (Azienda Ospedaliero-Universitaria Senese) Ethical Review Board, Siena, Italy (Prot n. 16929, dated March 16, 2020).

### Decision letter and Author response
Decision letter https://doi.org/10.7554/eLife.67569.sa1
Author response https://doi.org/10.7554/eLife.67569.sa2

## Additional files
### Supplementary files
• Reporting standard 1. STROBE checklist.
• Transparent reporting form

### Data availability
Sequencing data have been deposited in CINECA through http://www.nig.cineca.it/, specifically, http://nigdb.cineca.it., in the COVID-19 section through http://nigdb.cineca.it./registration/login.php. There are no restrictions on data access. Only registration is needed.

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
