## [Decision Letter]

**Acceptance summary:**

The authors provide solid evidence for the role of Toll-like receptor 7 in host defense against SARS Coronavirus-2. Based on the initial observation by Van der Made et al. (JAMA 324:1-22, 2020) that mutations in TLR-7 may lead to severe and even lethal COVID in young males, the authors found missense deleterious TLR-7 mutations in some 2 % of severe COVID male patients. In these patients there is a severe impairment of the Type-I and type-II interferon responses.

**Decision letter after peer review:**

Congratulations, we are pleased to inform you that your article, "Association of Toll-like receptor 7 variants with life-threatening COVID-19 disease in males", has been accepted for publication in *eLife*.

---

## [Author Response]

[Editors' note: we include below the reviews that the authors received from another journal, along with the authors’ responses.]

Editor's specific comments:

Please see the reviewers' comments below.

Reviewer #1: Major comments:

The authors should include a section on Statistical Methods that includes

a brief mention of Fisher's exact test for Table 1*a brief rationale for the use of LASSO logistic regression (with a reference**a brief explanation of why principal components analysis was applied prior to the LASSO logistic regression*

*a description of the cross-validation method and construction of the confusion matrix (with a reference)*

Added in the Online Repository file.

Reviewer #2:Fallerini et al. study TLR7 variants in males with mild compared with severe COVID19 infections in an Italian and Spanish cohort.

Comments

The methods suggest that 1,178 patients were included in the analysis, while it was 156 Italians and 122 Spanish. The fact that all were white European should be noted. Refine. The PBMC analysis of gene expression should also be noted.

A subset of 156 <60-year old male COVID-19 patients was selected from the Italian GENCOVID cohort of 1,178 SARS-CoV-2-infected subjects. We refined it in the text. We have now specified that all individuals were of European Caucasoid ethnicity in the Abstract and in the text as well as for PBMC analysis of gene expression.

Capsule summary – this section should be rewritten to highlight the key results in a quantitative format. Introductory statements should be removed.

Agreed and modified as suggested.

"strong predisposing factors". This statement should be toned down and be more precise as only 4% of the affected cohort had this variant.

As suggested by the reviewer, we have toned down the statement in the Capsule Summary.

Quantitate the relative risk of severe disease in males compared with females

More information has been provided to quantitate the relative risk of severe disease in males compared with females.

Reference 4. This reference is from 2004, when SARS-CoV-2 was not around. Amend.

Agreed and the reference replaced, updating the paragraph in the text.

The fact that details of methods are in an online repository should be stated.

Done.

5 out of 79 patients – add percentage

As suggested by the reviewer, the percentage (6.32%) has been added.

Round 0.0366 to 0.04.

Done.

Describe results in more detail/quantitatively

Agreed and rephrased as suggested.

Round 57.5 to 58 years

Done.

Table 1 – change "Marginal Row totals" to "total" in column and row headings. Round 0.0366 to 0.04.

Done (Table 1).

Table 2 – Describe in footer what is meant by "clinical category" 3 and 4.

Done (Table 2).

Figure 1 Refine some sentences in this figure legend focusing on the facts pertaining to the figure, without interpretation. Panel B currently comes after panel C – suggest reordering; or removing as it's significant is unclear.

Agreed and modified as suggested by the reviewer. More specifically, legend to Figure 1 is now more factual, and figure panels have been reordered and coordinated with the legend.

Figure 1 – Panel B might be removed. Panel E- it is unclear which line refers to which – revise.

As suggested by the reviewer, Figure 1 has been refined with the reordering of the panels and the exclusion of Panel E. Panel C (ex Panel B), reporting the confusion matrix, could be useful for the evaluation of the number of false negative/false positive of the classification.

Figure 2 – focus on comparison between affected patients and Ctl, not C1 – amend in all panels. For clarity, leave out comparison between C1 and patients – just comment in text.

We agree on the changes proposed for Figure 2 and modified statistics accordingly.

Figure 3 – table – reorder with generation I members first, then generation II members and finally generation III members. With females in the pedigree, clarify whether or not they required any hospital treatment.

As suggested by the reviewer, the generations in Figure 3 have been rearranged. We have specified in the text that females did not require hospital admission.

Reviewer #3:In this manuscript, the authors report a higher frequency of rare TLR7 variants in younger (<60 years) males with life-threatening COVID-19 than in a control group with asymptomatic or oligosymptomatic infection. PBMC from three patients with TLR7 variants and life-threatening disease, from one subject with TLR7 variant and oligosymptomatic infection and from 4 healthy controls were challenged in vitro with imiquimod (a TLR7 agonist), and impaired expression of IRF7 was demonstrated in PBMC from patients with life-threatening disease. The authors conclude that deleterious TLR7 variants may account for up to 4% of severe disease in male subjects.The study expands on a recent observation of two families in which COVID-19 segregated as an Xlinked recessive trait conditioned by SARS-CoV-2 infection.Overall, the study is interesting. However, some of the conclusions are overstated. Some methodological aspects need to be better defined. the organization of the manuscript should be improved, and reference to recent important findings by other groups on monogenic variants associated with life-threatening COVOD-19 must be added.Major comments:1) Some of the conclusions raised by the authors are overstated. In particular, in Figure 2, IRF7 and IFN- are the only transcripts that appear to be differentially expressed between patients with life threatening disease and healthy controls. For TLR7, this difference exists only between controls and P8, and for ISG15 between the healthy controls vs. P3 and P8.

As suggested by the reviewer, we have now tested all *TLR7* variants and modified the conclusions according to our recent findings. Thus we showed a significant impairment of TLR7 signalling pathway in the Ser301Pro, His630Tyr, and Ala1032Thr variants.

In the text, the authors emphasize the difference in the expression of these genes between patients with life-threatening disease and the oligosymptomatic SARS-CoV-2 infected patients, but this is not relevant if there is no difference versus healthy controls. Furthermore, there are technical and methodological weaknesses that need to be addressed. In particular, expression of TLR7 protein should be examined by flow cytometry.

We thank the reviewer for raising this important point. Indeed, we have partially replied above to this question by reviewer 2 and we modified Figure 2 according to his/her comment. It is important to emphasize that functional experiments were carried out in those patients from whom PBMC were available for further experiments. Expression of TLR7 protein has now been examined by flow cytometry in monocytes and B cells from patients and healthy controls showing that both expressed the TLR7 protein at the intracellular level.

To formally prove that these TLR7 variants are loss of function (or hylomorphic), transfection experiments should be performed in TLR7-deficient cells, and response to the TLR7 agonist should be examined.

We thank the reviewer for this suggestion. We believe that transfections are really important when primary cells cannot be retrieved from mutated patients. The availability of PBMC carrying the different *TLR7* variants identified in this study would make transfections redundant. Notably, in this revised version, we were able to expand the number of variants analyzed.

Finally, it is not known at what point in the course of the disease PBMC from the patients were collected.

PBMC from all patients were collected approximately 6 months after recovery.

An impaired response may also reflect the specific functional status of the cells in that particular moment of the infection. This is why the transfection experiments mentioned above are particularly important.

To evaluate the functional status of the cell, we stimulated PBMC from patients and healthy controls with the TLR4 agonist lipopolysaccharide (LPS). The intracellular production of IL6 was evaluated in monocytes. The frequencies of IL6+CD14+ cells were comparable in patients and healthy controls demonstrating that the cells of the patients were functionally active.

2) Important recent advances in the genetic basis of COVID-19 have been neglected, perhaps because the manuscript was submitted around the time when these discoveries were made publicly available. In any case, the recent description of deleterious variants in genes involved in type I IFN synthesis or signaling to these molecules (Zhang et al., Science 2020) should be cited and commented

As suggested by the reviewer, a proper section and the relative reference to the Zhang Q. paper has been added.

3) The manuscript suffers from some organizational deficiencies. Figure 1 is cited only once in the text, but it is composed of multiple panels which are not properly mentioned and commented. The legend to this figure reported first on panel A, then on panel C (before mentioning panel B).

We thank the reviewer for the suggestion. As also requested by reviewer 2, Figure 1 has been refined and panels reordered with the exclusion of ROC curves (Panel E). Legend to Figure 1 is now clearer and coherent with the order of panels.

Minor comments:

4) Table 2 reports on the Clinical Category of the patients, however no mention is made in the text in regard to how were the clinical categories defined

As also suggested by reviewer 2, we have added a footer to Table 2 carrying a detailed description of the clinical categories and of all abbreviations listed in the table.

5) Patients from Spain were included to expand the number of patients studied. Mention of approval from the local Institutional Review Board(s) is missing for this patient population.

We have now mentioned the Spanish Institutional Review Board approval in the Online Repository file and in the main text.

6) Segregation of the disease in the family of P6 is shown in Figure 3. However, these are only circumstantial supportive data (due to the fact that only few individuals from this family were infected with SARS-CoV-2). As such, the figure should be moved to Supplementary. If the authors insist on commenting on it, then data on X-chromosome inactivation in PBMC lineages from female carriers of the TLR7 variant should be provided.

In addition to the segregation of the disease in the family of P6, we have performed segregation analysis also in a further available pedigree (family of P3) confirming previous findings. As suggested by the reviewer, we have also provided a functional analysis for all TLR7-related genes in both families (Figure 3).

Reviewer #4:This study points to a possible risk factor of severe Covid-19 in males carrying variants of TLR7, thus confirming and potentially extending a previously published study (van der Made et al).A serious limitation is that only 2 variants (P2 and P3) have been functionally validated.

Thank you for this valuable comment. We have now functionally validated all *TLR7* variants (see above responses to other reviewers).

Data in Figure 2 for P1 do not convincingly show a functional effect of that particular variant (Val. 219 Ile).

We agree with the reviewer that the *TLR7* variant carried by patient P1 (Val219Ile) has a smaller functional impact suggesting a hypomorphic effect.

If no material is available for the other patients, it is feasible to express the variants in cell lines and to test them. An assay of interferon type I production will be overall more convincing.

We have now tested PBMC from 7 of 8 cases and, since the missing patient (P4) carried the same mutation of P5, we have now functionally validated all *TLR7* variants. IFN-ɑ (type IIFN) was analyzed as gene expression (Figure 2).

Other comments:

– There is an inconsistency in figures, i.e number of at risk cases 150 (table 1) or 156 (text)

The total number of severely affected males was 156 (as mentioned in the text). Among them, 150 subjects did not have mutations in the *TRL7* gene (Table 1).

– Were the 77 controls from the first cohort males ?

Yes, all SARS-CoV-2-infected subjects included in the analysis from both cohorts were males. We have now clarified this in the text.

– Figure 3 is actually anecdotal especially since the TLR7 variant present in this family has not been functionally validated

Segregation analysis was confirmed in two distinct pedigrees from Italy and Spain patients (P3 and P6) and also supported by functional analysis in all *TLR7*-related genes (Figure 3).

– The recent paper by Zhang Q. et al. on interferon I pathway variants as risk factors for severe Covid-19 should be cited (Science, 2020 (6515):abbd4570).

Done.

Response to second decision letter

Reviewer #2Thank you to the authors for addressing my previous queries. There remain some further comments that I suggest need addressing. Line numbers refer to clean untracked manuscript:Suggest being more specific regarding number / percentage (round all percentages to one decimal places) of patients in the Italian, Spanish and total cohort that had pathogenic TLR7 gene variants – e.g. Italian – 2/79 (2.5%) (not 5), Spanish – 1/77 (1.3%), total ?/272 (?%) – - not clear how many there were in the entire male cohort.

Done.

All percentages were rounded to one decimal place.

We have removed the numbers from the Abstract and explained better in the text: 3/156 (1,9%) pathogenic TLR7 gene variants in severely affected young males and 5/261 (1,9%) in the entire male case cohort, irrespective of age.

Two reported families is not a "fraction". What is meant by the "broader and complex host genome situation”. Suggest revising these conclusions to be more accurate and clear. Remove word "significantly".

Done.

Clinical implications – Revise to be more specific and focused "This new yet complex scenario" means little to the reader.

Done.

Capsular summary – what was the exact size of the total cohort studied?

The Italian young male cohort includes 156 patients and the Spanish one 122 patients. We have revised the sentence.

Percentages of male and female ICU admission and deaths are almost the same – add in significance levels for this and also hospitalization data. If not significant state this.

Done.

5 out of 79 patients – Table suggests that 2 of this cohort had pathogenic variants? Round percentages to one decimal place.

Done.

Were these "rare missense mutations" considered pathogenic? How many patients were there in the entire cohort – ?272?

Among the additional “rare missense mutations” found in the entire male cohort of 561 COVID-19 individuals (regardless of age), the one found in the cases has been shown to be LOF (p.Ala1032Thr) and the one found in the control has been shown to be neutral (p.Val222Asp). We have revised the sentence to make it clearer.

2% is rather small percentage of the total – suggest toning down "tip of the iceberg" and being more focused revising phrase "broader genome scenario".

Done.

Round percentage to one decimal place. 3/156 does not equate to figures given in results text above. Also does not seem to include the other two TLR7 variants found in the older males – were the variants in the older males considered pathogenic or just VUS? If the later, you might consider separating the analysis and conclusions to focus on males under 60?

Percentages were rounded to one decimal place. The two older males shared the same mutation (p.Ala1032Thr) that has been shown to be LOF. We have updated the text to make the paragraph clearer.

Table 1 – Suggest changing N of mutated patients to 3 and the terminology to pathogenic variants?

Table 1 refers to the statistical analysis of sequencing data done before functional studies on all variants.

Table 2 – Suggest listing just the 3 patients with pathogenic variants (a), or otherwise putting the VUS in a separate part of the table (b)

The table has been divided, grouping the LOF mutations together followed by the Hypo mutation and the neutral two.

Figure 3B – can you add data on the 3rd pedigree (2nd Italian family) with a pathogenic variant.

Done. We added the 3rd pedigree in Figure 3.

Reviewer #3:In the revised version of the manuscript, the authors have tones down some statements and corrected some errors as per the reviewers' recommendations. They have also added new data to address other comments, however far from clarifying the observations raised by the reviewers, these new data raise new important questions and fail to demonstrate internal consistency.Major comments:1) This reviewer had requested that the authors perform transfection experiments of TLR7 variants into TLR7 knock-out cells in order to demonstrate causality. The authors have argued that availability of patient PBMC is sufficient to address this point, as it allows functional testing. There are two problems with this. First, unless rescue experiments are performed in the patient cells, it is not possible to conclude that the functional effects are directly related to the TLR7 variants. Second, and more importantly, the TLR7 protein expression data produced by the authors in the response to reviewers (and cited as "data not shown in the text") are inconsistent with the mRNA data included in Figure 2 and 3. In particular, TLR7 mRNA expression was markedly reduced in P3, P6, P7 and P8 as compared to controls. However, TLR7 protein expression was no different in P6 and in controls. While for P7 one could conclude that TLR7 protein expression was reduced, no data are provided for P3 and P8. Although different experimental conditions were used to analyze TLR7 mRNA and protein expression, it is very hard to reconcile normal TLR7 protein expression, but markedly reduced mRNA expression, in P6. These data require a more robust experimental setting, and confirm the importance of using transfection experiments.

Regarding TLR7 expression in PBMCs, there was a misunderstanding. Figure 2 and Figure 3 refer to the mRNA fold change (activated/basal mRNA levels ratio) and not to absolute mRNA levels. Therefore, these results are not comparable with protein expression data.

Transfection experiments are usually requested when (i) patient cells are not available for every mutation presented; (ii) it is the first time that a gene is associated with a disorder. We have shown the effect of each variant in patient-specific cells and the gene has already been associated with the disease (ref. 8). Thus, functional analysis in patients’ and control PBMC represent a robust outcome to support our conclusions.

However, we have considered the request of the reviewer and, in this short time, we have performed transfection experiments for the variants expected to have a functional effect, cloning a dedicated TLR7 plasmid for each of them. PCR based site-directed mutagenesis was performed in pUNO-hTLR7 plasmid (Invivogen) to generate specific plasmids for the single variants. Transfection experiments were performed in HEK293 cells that do not express endogenous TLR7 (Chehadeh and Alkhabbaz 2013). Cells were maintained in DMEM supplemented with 10% FBS, 1% L-Glutamine and 1% pen/strep at 37°C with 5% CO_2_. Transient transfections were performed using Lipofectamine 2000 (Invitrogen) according to manufacturer’s protocol: 3x10^4^ cell/well were seeded the day before in 6 well plates, and then transfected with 2μg of DNA. Expression of TLR7 protein was examined by flow cytometry 24 hours after transfection, showing expression of TLR7 protein at the intracellular level in all cases (Figure 2B).

After 24 hours from transfection, the cells were stimulated in duplicate experiments with Imiquimod at 1μg/ml for 4 hours and then total RNA was extracted with RNeasy Mini Kit (Qiagen), according to manufacturer’s protocol. cDNA was synthesized from 1μg of total RNA using QuantiTect Reverse Transcription kit (Qiagen) according to the manufacturer’s instructions. We evaluated expression of IFN-ɑ in Imiquimod stimulated and unstimulated cells by qRT-PCR using the same assay described for PBMCs, confirming the results obtained in PBMCs (Figure 2C).

2) The segregation data shown in Figure 3 are not meaningful and do not provide substantial support to the authors' claims. In particular, for pedigree II, also males who did not inherit the TLR7 variant should be tested for IFN-a, ISG15 and IFN-g mRNA expression. Without this essential internal control, the data provided do not help. Incidentally, labels on the X-axis of all mRNA expression data in Figure 3 are misaligned.

Done. Figure 3 has been updated and X-axis labels aligned.

Minor comments:

1) Throughout the manuscript, the authors should avoid use of the word “mutation” and replace it with “variant” or “deleterious variant” as appropriate

Done

Response to third decision letter

Reviewer #2: Thank you for addressing most of my previous comments and suggestions. A few comments remain:1) In Abstract Results – detail number numerator/denominator (percentage) of pathological TLR7 variants found in the overall affected groups.

The requested detail has been added.

2) Authors should be able to calculate statistical significance of gender differences in the Stokes et al. paper themselves using online Chi-square calculator from the raw data in the paper – suggest amending sentence rather than say "even if they reported descriptive analyses without statistical comparisons"

Statistical significance has been calculated and the sentence has been modified

accordingly.

3) Percentages should be x.y, rather than x,y.

Done

4) In text and Table IIa – suggest removing details regarding gene variants that have no (neutral) functional/clinical significance and highlighting only predicted pathological variants, as non pathological variants are of no clinical relevance / not disease causing. Remove Table IIb. Figure legends will need to be adjusted accordingly.

We have removed Table 2B as requested.

However, we did not remove information on neutral variants since these variants were not previously published and we performed structural and functional analyses to validate their functionality. We thus feel that their characterization could be an added value to the paper.

5) Table I: add percentage affected in column (N. mutated patients). Suggest revising headings to "N. WT variants; N. pathological variants", rather than mutated patients.

Done

Reviewer #3:Minor comments: The authors have adequately revised the manuscript. They have also performed transfection experiments and tested experimentally the functional effects of the TLR7 variants identified. These are very important data that support the authors'; conclusions. Surprisingly, they have elected to show them only in the point-by-point reply to the reviewers. These data should be added to the main manuscript (as Supplementary data, if so needed), because they provide strong support to the authors' findings.

As suggested from the reviewer, we have added results of transfection experiments to the manuscript as panel B and C to new Figure 2. Text and figure legend have been modified accordingly. Experimental details have been added in the “online repository file”.